# The impact of wearing facemask on COPD patients: A protocol of a systematic review and meta-analysis

Xuwen Chen[1☉], Ibrahim Sani[1☉], Xiaoli Xia[2], Yi Li[3], Caiyun Li[1], Feiyan Yue[1], Xinhua Wang[1]*, Shisan Bao[1]*, Jingchun Fan[1]*

1 Center for Laboratory and Simulation Training, School of Public Health, Center for Evidence-Based Medicine, Gansu University of Chinese Medicine, Lanzhou, Gansu, China, 2 Department of Geriatrics, Affiliated Hospital of Gansu University of Chinese Medicine, Lanzhou, Gansu, China, 3 Department of Respiratory Cadres, Gansu Provincial People's Hospital, Lanzhou, Gansu, China

☉ These authors contributed equally to this work.
* Wangxinhua@163.com (XW); profbao@hotmail.com (SB); fan_jc@126.com (JF)

**Data Availability Statement:** No datasets were generated or analysed during the current study. All relevant data from this study will be made available upon study completion.

## Abstract

### Introduction

Chronic obstructive pulmonary disease (COPD) is a common, irreversible but preventable disease characterized by persistent respiratory symptoms. The mortality rate of COPD is predicted to reach 5.4 million by the year 2060. Despite its heavy burden on healthcare expenditure worldwide, only 15% of cases are medically identified. The potential benefits of facemask-wearing for COPD patients remain a topic of debate.

### Methods

We will conduct a systematic review of all randomized trials and non-randomized controlled trials to evaluate the impact of facemasks on COPD patients. Our review will be based on literature obtained through a comprehensive search strategy across multiple electronic databases, including the Cochrane Library, Embase, PubMed, Web of Science, the Chinese Biomedical Database (SinoMed), and China National Knowledge Infrastructure (CNKI), with no restrictions on language or date of publication. Two independent researchers will extract and assess all relevant data using pre-designed data extraction forms. The included studies will be assessed using the Cochrane RoB2 tool and the suggested risk of bias criteria proposed by the Effective Practice and Organization of Care reviews group of the Cochrane collaboration. The quality of evidence will be assessed using the Grading of Recommendations Assessment, Development and Evaluation (GRADE) approach. We will use Review Manager 5.4 software for statistical analysis.

### Discussion

In the context of COVID-19, it is important for COPD patients to wear facemasks. This study aims to conduct a comprehensive and systematic assessment of the impact of facemasks on the physiology and activity of COPD patients.

**Funding:** The study is supported by a grant from the National Key R & D Program "Precision Medicine Research"[2017YFC0907202], Gansu Provincial Administration of Traditional Chinese Medicine [grant no.GZK-2019-33], the 2020 Science and Technology Project of Chengguan District, Lanzhou [grant no.2020-2-11-16] and Natural Science Foundation Project of Gansu Province [grant no.22JR5RA589]. The funders had and will not have a role in study design, data collection and analysis, decision to publish, or preparation of the manuscript.

**Competing interests:** The authors have no conflicts of interest to declare.

**Trial registration: PROSPERO registration number** CRD42022326265.

## Introduction

Chronic obstructive pulmonary disease (COPD) is a common, irreversible but preventable disease characterized by persistent respiratory symptoms. The most common conditions of COPD are emphysema and chronic bronchitis, which cause irreversible damage [1]. Small airway disease occurs prior to emphysema, contributing to irreversible airway obstruction [2]. Common clinical features of COPD patients include shortness of breath, wheezing, and chronic cough [3]. COPD is associated with various risk factors, including smoking and exposure to air pollution caused by solid fuels and environmental particulate matter [4]. The global prevalence of COPD was 10.3% among people aged 30–79 in 2019 [5]. COPD is the fourth or third leading cause of global disability-adjusted life years (DALYs) affecting people aged 50–74 or over 75 years, respectively [6]. It is predicted that the mortality rate of COPD will exceed 5.4 million by 2060 [7].

Since the COVID-19 pandemic, facemasks have received more attention [8]. As personal protective equipment, facemasks can prevent particulate matter, allergens, and pathogens from entering the respiratory system [9]. Common types of facemasks include disposable respirators, surgical masks, and cloth masks [10]. It is well known that wearing facemasks can reduce the risk of respiratory pathogen infection and transmission [11,12]. Moreover, facemasks can effectively reduce the intake of air pollution particles, providing protection against air pollution [13,14].

Studies have found that air pollution and respiratory infections are risk factors for exacerbation of COPD [15,16]. According to the GOLD 2023 committee report, there is a supra-linear relationship between air pollution and respiratory events, indicating that there is no absolutely safe level of air pollution [17]. Wearing a facemask would benefit the general population, particularly COPD patients who are more vulnerable to invisible pollution and could experience acute exacerbations [18]. Therefore, it is advisable for COPD patients to wear masks in normal and/or specific environments to minimize disease progression [19].

However, wearing facemasks can also increase inspiratory and expiratory airflow resistance, resulting in shortness of breath and reduced exercise time for some individuals, including those with COPD [20]. To date, no systematic review or meta-analysis has evaluated the potential impact of wearing facemasks on COPD patients. Therefore, the current study aims to determine the impact of wearing facemasks on physiological indexes and activities in patients with COPD through a systematic review and meta-analysis. The findings of this study will provide a reference for clinical guidance on wearing facemasks during activities for COPD patients.

## Methods

### Study registration and protocol

This study was registered on the International Prospective Register of Systematic Reviews (PROSPERO) in April 2022. The registration number is CRD42022326265. The Preferred Reporting Items for Systematic Reviews and Meta-Analysis (PRISMA) will be used to report the systematic review and meta-analysis [21].

## Criteria for inclusion of studies in the review

**Types of studies.**   Randomized trials (including randomized crossover trials) and non-randomized controlled trials are eligible for inclusion. Although we will search the database for published outcomes, the experiment design we intend to include is a prospective study that comprises prospective intervention trials for both randomized and non-randomized controlled trials.

**Types of participants.**   We will include patients who have been clinically diagnosed with COPD, as defined by the investigators, regardless of age, sex, and severity of the disease.

**Types of interventions.**   The interventions we will consider include surgical masks, N95 masks, and all other types of masks not used for treatment purposes, regardless of the duration and frequency of wearing.

**Types of comparators.**   The systematic review and meta-analysis will encompass two distinct comparisons: 1. Comparisons involving COPD patients in the mask group and those in the non-mask group. 2. Comparisons within COPD patients who wear different types of masks.

**Types of outcomes.**   The primary outcomes of this study will include end-tidal carbon dioxide ($ETCO_2$), heart rate (HR), respiratory rate (RR), and oxygen saturation ($SpO_2$). The secondary outcomes will involve other physiological variables and indicators related to exercise capacity, such as minute ventilation, inspiratory time, blood pressure, pulmonary function, blood lactate, oxygen partial pressure, carbon dioxide partial pressure, 6-minute walking distance, exercise performance capacity, and work rate.

It is worth mentioning that, based on preliminary search and published data, we identified several trials of mask-related interventions involving $ETCO_2$ [22–24]. Similarly, the protocol anticipates the importance of $ETCO_2$, as an indicator that can be obtained through non-invasive technology, for assessing carbon dioxide levels in mask-wearing COPD patients [25].

## Exclusion criteria

1. Animal studies.

2. Other systematic reviews and meta-analyses.

3. Studies without complete data.

4. Full texts are not available.

5. The facemasks involved in the study were used as interfaces for non-invasive positive-pressure ventilation, including nasal masks and oronasal masks.

6. Duplicate publications or overlapping studies.

## Information sources and search strategy

We will use a comprehensive search strategy to select eligible studies from multiple electronic databases, including the Cochrane Library, Embase, PubMed, Web of Science, the Chinese Biomedical Database (SinoMed), and China National Knowledge Infrastructure (CNKI), with no restrictions on language or date of publication. An example search strategy for the PubMed database is described in Table 1 below. References of all relevant articles were retrieved to further search for eligibility.

## Study selection

We will import all retrieved documents into Endnote X9 software for filtering and record the entire process of our primary and secondary screening. After removing duplicates, two

**Table 1. Search strategy (PubMed).**

| |
|---|
| #1 Search "Pulmonary Disease, Chronic Obstructive" [Mesh] |
| #2 Search (((((((Chronic Obstructive Lung Disease [Title/Abstract]) OR (Chronic Obstructive Pulmonary Diseases [Title/Abstract])) OR (COAD [Title/Abstract])) OR (COPD [Title/Abstract])) OR (Chronic Obstructive Airway Disease [Title/Abstract])) OR (Chronic Obstructive Pulmonary Disease [Title/Abstract])) OR (Chronic Airflow Obstructions [Title/Abstract])) OR (Chronic Airflow Obstruction [Title/Abstract]) |
| #3 Search #1 OR #2 |
| #4 Search "Masks" [Mesh] OR "N95 Respirators" [Mesh] |
| #5 Search (((((mask [Title/Abstract]) OR (masks [Title/Abstract]))OR (facemask [Title/Abstract])) OR (facemasks [Title/Abstract])) OR (face-mask [Title/Abstract]))OR(face-masks [Title/Abstract]) OR (((((((N95 Respirator [Title/Abstract]) OR (N95 Face Masks [Title/Abstract])) OR (N95 Face Mask [Title/Abstract])) OR (N95 Masks [Title/Abstract])) OR (N95 Mask [Title/Abstract])) OR (N95 Filtering Facepiece Respirators [Title/Abstract])) OR (N95 FFRs [Title/Abstract])) OR (N95 FFR [Title/Abstract]) |
| #6 Search #4 OR #5 |
| #7 Search #3 AND #6 |

reviewers will independently screen all abstracts and titles to exclude irrelevant reports. We will download and read all relevant publications that meet our selection criteria in full text. The results will be discussed by the two reviewers, and disagreements will be resolved by consulting with a third reviewer to reach a consensus.

## Data extraction

Two researchers will independently extract all relevant data using pre-designed data extraction forms, including titles, authors, journals, research time, participants' demographics, research methods, and results from measurements. We will resolve any discrepancies through discussion and consult a third senior researcher if necessary.

## Risk of bias assessment

Two assessors will independently evaluate each study, and any disagreement will be resolved through a consensus discussion. The Cochrane RoB2 tool will be used to assess randomized controlled trials and randomized crossover trials [26]. This tool will address five specific domains: randomization process, deviations from intended intervention, missing outcome data, measurement of outcomes, and selection of the reported result. Other study designs will be assessed using the suggested risk of bias criteria proposed by the Effective Practice and Organization of Care reviews group of the Cochrane collaboration [27]. Non-randomized controlled trials will be assessed for nine risk-of-bias domains, including random sequence generation, allocation concealment, baseline outcome measurements similarity, baseline characteristics similarity, incomplete outcome data, prevention of knowledge of the allocated interventions during the study, protection against contamination, selective outcome reporting, and other risks of bias.

## Statistical analysis

Statistical analysis will be performed using Review Manager 5.4 software provided by the Cochrane Collaboration. Since the outcome indicators, we intend to collect are continuous results. Mean Difference (MD) will be used as the effect indicator. Standardized mean difference (SMD) will be used as the effect indicator if the measurement tools and measurement units are different. Results will be presented with 95% confidence intervals (CIs).

For continuous variables, we will record the baseline value and final value after the intervention for each group, including means and standard deviations. These data will be extracted

from tables or text or calculated from other known data when necessary. For trials lacking the mean and SD of the final value, we will use formulas to impute relevant information based on the Cochrane Collaboration Handbook guidelines [28], assuming a correlation coefficient (Corr) of 0.5 [28,29].

$$Mean_{change} = Mean_{final} - Mean_{baseline}$$

$$SD_{change} = \sqrt{SD_{baseline}^2 + SD_{final}^2 - 2 \times Corr \times SD_{baseline} \times SD_{final}}$$

If relevant data are not reported, we will contact the researcher or study sponsor by email to request the missing data, especially for data needed for meta-analysis. When this is not possible, currently available data will be analyzed and discussed as limitations.

Heterogeneity across included studies will be assessed using the Q statistic and I-square index. We will use the fixed-effect model for homogeneous studies and the random-effects model for others.

## Subgroup analysis

If sufficient data are extracted, we will conduct subgroup analysis in the following two aspects.

1. Participant characteristics, including sex, age, disease severity, vaccination history, medication history, educational background, caregiver support at home, and other relevant factors.

2. Different types of facemasks: surgical masks, N95 masks, cloth masks, etc.

3. Characterization of mask wear: duration of mask wear and frequency of mask wear.

## Sensitivity analysis

We also plan to remove the effect of studies with a high risk of bias for sensitivity analysis to consider the difference in pooled effects.

## Assessment of publication bias

When a meta-analysis contains 10 or more studies, we plan to create funnel plots to assess publication bias.

## Evidence quality evaluation

We will use the Grading of Recommendations Assessment, Development and Evaluation (GRADE) approach to assess the quality of evidence for the outcomes of the meta-analysis. The GRADE method downgrades the evidence based on five aspects, including study limitations, inconsistency of results, indirectness of evidence, imprecision, and publication bias. The GRADE approach categorizes the quality of evidence as high, moderate, low, or very low [30]. We will use the GRADEpro GDT online tool for this assessment.

## Amendments

If we need to modify the protocol, we will describe the change and give reasons in the final study.

## Discussion

Since the SARS-CoV-2 pandemic, many countries have recommended wearing facemasks to prevent pathogen transmission [31], leading to increased attention and research on facemasks. Currently, most systematic reviews on facemasks include healthy individuals [32], with limited research on COPD patients. COPD is a chronic and irreversible airway disease with significant personal and social impacts [33]. Therefore, this study aims to conduct a comprehensive and systematic assessment of the impact of wearing facemasks on COPD patients, guided by PRISMA [34], and provide objective evidence for future research. As the first systematic review and meta-analysis to assess the impact of facemasks on COPD patients, we believe that this study has important clinical and practical significance, considering the current situation and the reality of patients.

This is a protocol for investigating the potential impact of wearing masks among COPD patients. To ensure feasibility, this protocol plans to search without limiting the characteristics of patients with COPD, the type of mask, and the frequency and duration of mask wearing to minimize the loss of studies related to mask wearing in patients with COPD. In addition, to ensure that we find as many studies as possible, there are no language, year, or country restrictions in this protocol, and data will be collected by searching six databases, including two Chinese-language databases. Based on our preliminary search and screening results and published data, we have identified over 30 studies relating to mask wearing in patients with COPD. Thus, our search strategy, inclusion criteria, and selection of databases can ensure that a systematic review is feasible.

This study has some limitations. Firstly, we will only include and evaluate published research, and unpublished literature will not be considered. Secondly, the number of papers related to COPD and facemasks is relatively small, making it challenging to design randomized controlled trials with facemasks for COPD patients, which may result in inconclusive outcomes from this study.

## Supporting information

**S1 File. PRISMA-P 2015 checklist.**
(DOCX)

## Acknowledgments

The authors are grateful to all the experts and research institutions who have helped us with our research.

## Author Contributions

**Conceptualization:** Xiaoli Xia, Yi Li, Xinhua Wang, Jingchun Fan.

**Data curation:** Ibrahim Sani, Caiyun Li, Feiyan Yue.

**Formal analysis:** Xuwen Chen, Ibrahim Sani.

**Methodology:** Xuwen Chen, Shisan Bao, Jingchun Fan.

**Writing – original draft:** Xuwen Chen, Ibrahim Sani.

**Writing – review & editing:** Shisan Bao, Jingchun Fan.

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
