## [Decision Letter · Decision Letter 0]

13 Jul 2023

PONE-D-23-09305The impact of wearing facemask on COPD patients: A protocol of a systematic review and meta-analysisPLOS ONE

Dear Dr. Fan,

Thank you for submitting your manuscript to PLOS ONE. After careful consideration, we feel that it has merit but does not fully meet PLOS ONE’s publication criteria as it currently stands. Therefore, we invite you to submit a revised version of the manuscript that addresses the points raised during the review process.

We look forward to receiving your revised manuscript.

Kind regards,

Dong Keon Yon, MD, FACAAI, FAAAAI

Academic Editor

PLOS ONE

Additional Editor Comments:

This is an excellent paper. Please address the excellent comments from the reviewers.

Ref 18, 19 -> Please updated PRISMA guideline (2015 -> 2020); DOI: https://doi.org/10.54724/lc.2022.e9

Thank you.

Reviewers' comments:

Reviewer's Responses to Questions

**Comments to the Author**

1. Does the manuscript provide a valid rationale for the proposed study, with clearly identified and justified research questions?

Reviewer #1: Partly

Reviewer #2: Partly

Reviewer #3: Yes

Reviewer #4: No

2. Is the protocol technically sound and planned in a manner that will lead to a meaningful outcome and allow testing the stated hypotheses?

Reviewer #1: Partly

Reviewer #2: No

Reviewer #3: Partly

Reviewer #4: No

3. Is the methodology feasible and described in sufficient detail to allow the work to be replicable?

Reviewer #1: Yes

Reviewer #2: No

Reviewer #3: Yes

Reviewer #4: No

4. Have the authors described where all data underlying the findings will be made available when the study is complete?

Reviewer #1: No

Reviewer #2: Yes

Reviewer #3: No

Reviewer #4: No

5. Is the manuscript presented in an intelligible fashion and written in standard English?

Reviewer #1: Yes

Reviewer #2: No

Reviewer #3: Yes

Reviewer #4: No

6. Review Comments to the Author

You may also provide optional suggestions and comments to authors that they might find helpful in planning their study.

Reviewer #1: The protocol seems to be interesting but , I have a few questions for the authors.

1) One group is the COPD patients who are wearing masks and the other group comprises of normal people/ COPD patients who are not wearing masks?

2) Are you taking into account the vaccination history of these patients like influenza or the pneumococcal because they could have an impact on the number of exacerbations?

3) What about the medication history of the patients including oral and inhaled steroids?

4) The educational background of the individual and the caregiver support at home is also an important factor to be considered. Is it being done in this study?

Thank you.

Reviewer #2: The study design appears retrospective, looking at databases. Yet it implies intervention in terms of mask wear. Further more, it implies checking at EtCO2, 6MWT at end of study. Intervention and prospective assessment cannot be done in a retrospective study.

Reviewer #3: Authors are required to present the plan of data availability in respective question. Although data is not generated yet even then a data sharing plan may be given as accorded by the Plos One policy.

Reviewer #4: This manuscript is a proposal of potential meta-analysis about the impact of wearing facemask on the clinical outcomes of COPD patients. I have below comments.

1. It’s not clear about the importance of this topic about the correlation between wearing facemask and the clinical outcomes of COPD patients. If the environment and air quality is good, facemask would be unnecessary for COPD patients.

2. It is not clear how many published studies are available for meta-analysis to summarize the correlation between wearing facemask and the clinical outcomes of COPD patients. Authors should have provided such information to confirm the feasibility of the proposed systemic review.

3. Lines 126-128, Types of interventions may not be comparable relative to the clinical outcome if duration and frequency of wearing facemask are not considered.

4. It is not clear what mean differences will be collected and summarized. Types of studies include randomized and non-randomized trials, while the proposed statistical analysis will examine mean change at final from mean at baseline. How are these means obtained from randomized and non-randomized trials if their designs are not looking at baseline and final?

5. Line 209 – 214, if trials don’t provide mean and SD of the final, the proposed formula is not useful.

6. Please look for help from English editor to the writing.

7. PLOS authors have the option to publish the peer review history of their article (what does this mean?). If published, this will include your full peer review and any attached files.

Reviewer #1: No

Reviewer #2: No

Reviewer #3: **Yes: **Muhammad Kashif Munir PhD

Reviewer #4: No

---

## [Author Response · Author response to Decision Letter 0]

1 Aug 2023

Dr Dong Keon Yon 

Academic Editor

PLOS ONE

1 Aug 2023

Dear Dr Yon

We appreciate the constructive comments made by the reviewers, and our responses are as follows:

Comments to the Author

1. Does the manuscript provide a valid rationale for the proposed study, with clearly identified and justified research questions?

Reviewer #1: Partly

Reviewer #2: Partly

Reviewer #3: Yes

Reviewer #4: No

We have made corrections and edited our manuscript accordingly.

2. Is the protocol technically sound and planned in a manner that will lead to a meaningful outcome and allow testing the stated hypotheses?

Reviewer #1: Partly

Reviewer #2: No

Reviewer #3: Partly

Reviewer #4: No

We have made corrections and edited our manuscript accordingly.

3. Is the methodology feasible and described in sufficient detail to allow the work to be replicable?

Reviewer #1: Yes

Reviewer #2: No

Reviewer #3: Yes

Reviewer #4: No

We have made corrections and edited our manuscript accordingly.

4. Have the authors described where all data underlying the findings will be made available when the study is complete?

Reviewer #1: No

Reviewer #2: Yes

Reviewer #3: No

Reviewer #4: No

We have modified our manuscript in the “Data Availability Statement”. (Data Availability Statement, lines 304-306, paragraph 2 page 16)

5. Is the manuscript presented in an intelligible fashion and written in standard English?

Reviewer #1: Yes

Reviewer #2: No

Reviewer #3: Yes

Reviewer #4: No

Thanks. This manuscript has been proofread by a native English speaker with scientific background.

6. Review Comments to the Author

Reviewer #1: The protocol seems to be interesting but, I have a few questions for the authors.

1. One group is the COPD patients who are wearing masks and the other group comprises of normal people/ COPD patients who are not wearing masks?

During the study period, the trial group of COPD patients wore masks, while the other group served as the control and either did not wear facemasks or wore different facemasks. To clarify this point, we have modified our manuscript accordingly. It now reads: “We will include COPD patients who did not wear facemasks or wore different facemasks as controls, compared to the experimental group who wore masks during the study period.” (Methods, line 130-132 paragraph 2 page 7) 

2. Are you taking into account the vaccination history of these patients like influenza or the pneumococcal because they could have an impact on the number of exacerbations?

3. What about the medication history of the patients including oral and inhaled steroids?

4. The educational background of the individual and the caregiver support at home is also an important factor to be considered. Is it being done in this study?

We believe that points 2, 3, and 4 are closely related, and our response is as follows:

To clarify this point, we have modified our manuscript to include the following: “1. Participant characteristics, including sex, age, disease severity, vaccination history, medication history, educational background, caregiver support at home, and other relevant factors.” (Methods, line 230-232 paragraph 2 page 12)

Reviewer #2: The study design appears retrospective, looking at databases. Yet it implies intervention in terms of mask wear. Furthermore, it implies checking at EtCO2, 6MWT at end of study. Intervention and prospective assessment cannot be done in a retrospective study.

We apologize for any confusion. To clarify, we have added the following sentences to the manuscript: “Although we will search the database for published outcomes, the experiment design we intend to include is a prospective study that comprises prospective intervention trials for both randomized and non-randomized controlled trials.” (Methods, line 115-118 paragraph 2 page 6) 

Reviewer #3: Authors are required to present the plan of data availability in respective question. Although data is not generated yet even then a data sharing plan may be given as accorded by the Plos One policy.

We have modified our manuscript in the “Data Availability Statement”. (Data Availability Statement, lines 304-306, paragraph 2 page 16)

Reviewer #4: This manuscript is a proposal of potential meta-analysis about the impact of wearing facemask on the clinical outcomes of COPD patients. I have below comments.

1. It’s not clear about the importance of this topic about the correlation between wearing facemask and the clinical outcomes of COPD patients. If the environment and air quality is good, facemask would be unnecessary for COPD patients.

To clarify this point, we have added the following sentences to the manuscript: “According to the GOLD 2023 committee report, there is a supra-linear relationship between air pollution and respiratory events, indicating that there is no absolutely safe level of air pollution [17]. Wearing a facemask would benefit the general population, particularly COPD patients who are more vulnerable to invisible pollution and could experience acute exacerbations [18]. Therefore, it is advisable for COPD patients to wear masks in normal and/or specific environments to minimize disease progression [19].” (Introduction, line 88-94 paragraph 1 page 5)

17.Sin DD, Doiron D, Agusti A, Anzueto A, Barnes PJ, Celli BR, et al. Air pollution and COPD: GOLD 2023 committee report. Eur Respir J. 2023; 61(5):2202469. https://doi:10.1183/13993003.02469-2022 PMID: 36958741

18.Jiang XQ, Mei XD, Feng D. Air pollution and chronic airway diseases: what should people know and do? J Thorac Dis. 2016; 8(1):E31-40. https://doi:10.3978/j.issn.2072-1439.2015.11.50 PMID: 26904251

19.Kyung SY, Jeong SH. Particulate-Matter Related Respiratory Diseases. Tuberc Respir Dis (Seoul). 2020; 83(2):116-21. https://doi: 10.4046/trd.2019.0025 PMID: 32185911

2. It is not clear how many published studies are available for meta-analysis to summarize the correlation between wearing facemask and the clinical outcomes of COPD patients. Authors should have provided such information to confirm the feasibility of the proposed systemic review.

To clarify this point, we have added the following sentences to the manuscript: “This is a protocol for investigating the potential impact of wearing masks among COPD patients. To ensure feasibility, this protocol plans to search without limiting the characteristics of patients with COPD, the type of mask, and the frequency and duration of mask wearing to minimize the loss of studies related to mask wearing in patients with COPD. In addition, to ensure that we find as many studies as possible, there are no language, year, or country restrictions in this protocol, and data will be collected by searching six databases, including two Chinese-language databases. Based on our preliminary search and screening results and published data, we have identified over 30 studies relating to mask wearing in patients with COPD. Thus, our search strategy, inclusion criteria, and selection of databases can ensure that a systematic review is feasible.” (Discussion, lines 270-280, paragraph 2 page 14-15)

3. Lines 126-128, Types of interventions may not be comparable relative to the clinical outcome if duration and frequency of wearing facemask are not considered.

To clarify this point, we have added the following sentences accordingly in the Methods section, it now reads: “2. Different types of facemasks: surgical masks, N95 masks, cloth masks, etc. 3. Characterization of mask wear: duration of mask wear and frequency of mask wear.” (Methods, line 234 paragraph 2 page 12)

4. It is not clear what mean differences will be collected and summarized. Types of studies include randomized and non-randomized trials, while the proposed statistical analysis will examine mean change at final from mean at baseline. How are these means obtained from randomized and non-randomized trials if their designs are not looking at baseline and final?

To clarify this point, we have modified our manuscript as follows: “For continuous variables, we will record the baseline value and final value after the intervention for each group, including means and standard deviations. These data will be extracted from tables or text or calculated from other known data when necessary. For trials lacking the mean and SD of the final value, we will use formulas to impute relevant information based on the Cochrane Collaboration Handbook guidelines [25], assuming a correlation coefficient (Corr) of 0.5 [25, 26].” 

〖Mean〗_change=〖Mean〗_final-〖Mean〗_baseline

〖SD〗_change=√(〖SD〗_baseline^2+〖SD〗_final^2-2×Corr×〖SD〗_baseline×〖SD〗_final )

If relevant data are not reported, we will contact the researcher or study sponsor by email to request the missing data, especially for data needed for meta-analysis. When this is not possible, currently available data will be analyzed and discussed as limitations. Heterogeneity across included studies will be assessed using the Q statistic and I-square index. We will use the fixed-effect model for homogeneous studies and the random-effects model for others.” (Methods, line 210-221 paragraph 2 page 11)

5. Line 209 – 214, if trials don’t provide mean and SD of the final, the proposed formula is not useful.

As mentioned above, the formula can be used to calculate the final value when the included studies provide the baseline value and the change from baseline value. When final values and baseline values are not given, we will attempt to obtain the data by contacting the researcher or study sponsor.

6. Please look for help from English editor to the writing.

This manuscript has been proofread by a native English speaker with scientific background.

We have revised our manuscript accordingly, and hope our manuscript meet the standard for publication in PLOS ONE.

Yours sincerely

Jingchun Fan

---

## [Decision Letter · Decision Letter 1]

21 Aug 2023

PONE-D-23-09305R1The impact of wearing facemask on COPD patients: A protocol of a systematic review and meta-analysisPLOS ONE

Dear Dr. Fan,

Thank you for submitting your manuscript to PLOS ONE. After careful consideration, we feel that it has merit but does not fully meet PLOS ONE’s publication criteria as it currently stands. Therefore, we invite you to submit a revised version of the manuscript that addresses the points raised during the review process.

We look forward to receiving your revised manuscript.

Kind regards,

Dong Keon Yon, MD, FACAAI, FAAAAI

Academic Editor

PLOS ONE

Journal Requirements:

Additional Editor Comments:

This is an excellent paper. Please address the excellent comments from the reviewers.

And Please address my minor comment.

Ref 18, 19 -> Please updated PRISMA guideline (2015 -> 2020); DOI: https://doi.org/10.54724/lc.2022.e9

Thank you.

Reviewers' comments:

Reviewer's Responses to Questions

**Comments to the Author**

1. Does the manuscript provide a valid rationale for the proposed study, with clearly identified and justified research questions?

Reviewer #1: Yes

Reviewer #2: Partly

Reviewer #4: Yes

2. Is the protocol technically sound and planned in a manner that will lead to a meaningful outcome and allow testing the stated hypotheses?

Reviewer #1: Yes

Reviewer #2: No

Reviewer #4: Yes

3. Is the methodology feasible and described in sufficient detail to allow the work to be replicable?

Reviewer #1: Yes

Reviewer #2: Yes

Reviewer #4: Yes

4. Have the authors described where all data underlying the findings will be made available when the study is complete?

Reviewer #1: Yes

Reviewer #2: Yes

Reviewer #4: No

5. Is the manuscript presented in an intelligible fashion and written in standard English?

Reviewer #1: Yes

Reviewer #2: Yes

Reviewer #4: Yes

6. Review Comments to the Author

You may also provide optional suggestions and comments to authors that they might find helpful in planning their study.

Reviewer #1: all comments have been addressed to satisfaction and the article may be accepted if found to be suitable by the editorial board.

Reviewer #2: The methodology is still confusing. Study group is mask wearing, while control group is other masks wearing / no mask wearing?

EtCO2 is one of the parameters due for checking. However it is not routinely done in COPD patients.

Reviewer #4: All my comments have been addressed. No further comments.

The following is my previous comments.

This manuscript is a proposal of potential meta-analysis about the impact of wearing facemask on the clinical outcomes of COPD patients. I have below comments.

1. It’s not clear about the importance of this topic about the correlation between wearing facemask and the clinical outcomes of COPD patients. If the environment and air quality is good, facemask would be unnecessary for COPD patients.

2. It is not clear how many published studies are available for meta-analysis to summarize the correlation between wearing facemask and the clinical outcomes of COPD patients. Authors should have provided such information to confirm the feasibility of the proposed systemic review.

3. Lines 126-128, Types of interventions may not be comparable relative to the clinical outcome if duration and frequency of wearing facemask are not considered.

4. It is not clear what mean differences will be collected and summarized. Types of studies include randomized and non-randomized trials, while the proposed statistical analysis will examine mean change at final from mean at baseline. How are these means obtained from randomized and non-randomized trials if their designs are not looking at baseline and final?

5. Line 209 – 214, if trials don’t provide mean and SD of the final, the proposed formula is not useful.

6. Please look for help from English editor to the writing.

7. PLOS authors have the option to publish the peer review history of their article (what does this mean?). If published, this will include your full peer review and any attached files.

Reviewer #1: No

Reviewer #2: No

Reviewer #4: No

---

## [Author Response · Author response to Decision Letter 1]

11 Sep 2023

Dr Dong Keon Yon 

Academic Editor

PLOS ONE

11 September 2023

Dear Dr Yon

We appreciate the constructive comments made by the reviewers, and our responses are as follows:

Journal Requirements:

We confirm that there is no retracted reference cited in our manuscript.

Additional Editor Comments:

Thanks for the positive comment: “This is an excellent paper.”

Please address my minor comment.

Ref 18, 19 -> Please updated PRISMA guideline (2015 -> 2020); DOI: https://doi.org/10.54724/lc.2022.e9

We have modified our manuscript accordingly. It now reads: “The Preferred Reporting Items for Systematic Reviews and Meta-Analysis (PRISMA) will be used to report the systematic review and meta-analysis [21]” (Methods, lines 109-110, paragraph 1 page 6)

21. Lee S W, Koo M J. PRISMA 2020 statement and guidelines for systematic review and meta-analysis articles, and their underlying mathematics: Life Cycle Committee Recommendations. Life Cycle. 2022; 2:e9. https://doi.org/10.54724/lc.2022.e9

Comments to the Author

1. Does the manuscript provide a valid rationale for the proposed study, with clearly identified and justified research questions?

Reviewer #1: Yes

Reviewer #2: Partly

Reviewer #4: Yes

Thank you.

2. Is the protocol technically sound and planned in a manner that will lead to a meaningful outcome and allow testing the stated hypotheses?

Reviewer #1: Yes

Reviewer #2: No

Reviewer #4: Yes

We have modified our manuscript accordingly.

3. Is the methodology feasible and described in sufficient detail to allow the work to be replicable?

Reviewer #1: Yes

Reviewer #2: Yes

Reviewer #4: Yes

Thank you.

4. Have the authors described where all data underlying the findings will be made available when the study is complete?

Reviewer #1: Yes

Reviewer #2: Yes

Reviewer #4: No

We have modified our manuscript in the “Data Availability Statement”, it now reads: “No datasets were generated or analysed during the current study. All relevant data from this study will be made available upon study completion. Upon publication of the study results, a fully dataset will be made publicly available if required by the scientific journal in which the results are published. If the scientific journal does not require a full dataset, a fully dataset will be made available from the corresponding author upon reasonable request.” (Data Availability Statement, lines 306-310, paragraph 2 page 16)

5. Is the manuscript presented in an intelligible fashion and written in standard English?

Reviewer #1: Yes

Reviewer #2: Yes

Reviewer #4: Yes

Thank you.

6. Review Comments to the Author

Reviewer #1: all comments have been addressed to satisfaction and the article may be accepted if found to be suitable by the editorial board.

Reviewer #2: The methodology is still confusing. Study group is mask wearing, while control group is other masks wearing / no mask wearing?

We apologize for any confusion. To clarify this point, we have modified our manuscript accordingly. It now reads: “The systematic review and meta-analysis will encompass two distinct comparisons: 1. Comparisons involving COPD patients in the mask group and those in the non-mask group. 2. Comparisons within COPD patients who wear different types of masks.” (Methods, lines 130-132, paragraph 2 page 7) 

EtCO2 is one of the parameters due for checking. However, it is not routinely done in COPD patients.

Although ETCO2 is not routinely measured in patients with COPD, it is important as an indicator that can be obtained by non-invasive technology to assess CO2 levels in mask wearers. In addition, based on our preliminary search and published data, we have identified multiple interventional trials related to masks that have involved ETCO2.Therefore, we hypothesize that ETCO2 might also be measured in interventional trials of mask wearing in patients with COPD. To clarify this point, we have added the following sentences to the manuscript: “It is worth mentioning that, based on preliminary search and published data, we identified several trials of mask-related interventions involving ETCO2 [22-24]. Similarly, the protocol anticipates the importance of ETCO2, as an indicator that can be obtained through non-invasive technology, for assessing carbon dioxide levels in mask-wearing COPD patients [25].” (Methods, lines 142-146, paragraph 4 page 7)

22. Epstein D, Korytny A, Isenberg Y, Marcusohn E, Zukermann R, Bishop B, et al. Return to training in the COVID-19 era: The physiological effects of face masks during exercise. Scand J Med Sci Sports. 2021; 31(1):70-5. http://doi:10.1111/sms.13832 PMID: 32969531 

23. Lubrano R, Bloise S, Marcellino A, Ciolli CP, Testa A, De Luca E, et al. Effects of N95 Mask Use on Pulmonary Function in Children. J Pediatr. 2021; 237:143-7. http://doi:10.1016/j.jpeds.2021.05.050 PMID: 34043996

24. Bar-On O, Gendler Y, Stafler P, Levine H, Steuer G, Shmueli E, et al. Effects of Wearing Facemasks During Brisk Walks: A COVID-19 Dilemma. J Am Board Fam Med. 2021; 34(4):798-801. http://doi:10.3122/jabfm.2021.04.200559 PMID: 34312270

25. Brooks JP, Layman J, Willis J. Physiologic effects of surgical masking in children versus adults. PeerJ. 2023; 11:e15474. http://doi:10.7717/peerj.15474 PMID: 37342359

Reviewer #4: All my comments have been addressed. No further comments.

The following is my previous comments.

This manuscript is a proposal of potential meta-analysis about the impact of wearing facemask on the clinical outcomes of COPD patients. I have below comments.

1. It’s not clear about the importance of this topic about the correlation between wearing facemask and the clinical outcomes of COPD patients. If the environment and air quality is good, facemask would be unnecessary for COPD patients.

2. It is not clear how many published studies are available for meta-analysis to summarize the correlation between wearing facemask and the clinical outcomes of COPD patients. Authors should have provided such information to confirm the feasibility of the proposed systemic review.

3. Lines 126-128, Types of interventions may not be comparable relative to the clinical outcome if duration and frequency of wearing facemask are not considered.

4. It is not clear what mean differences will be collected and summarized. Types of studies include randomized and non-randomized trials, while the proposed statistical analysis will examine mean change at final from mean at baseline. How are these means obtained from randomized and non-randomized trials if their designs are not looking at baseline and final?

5. Line 209 – 214, if trials don’t provide mean and SD of the final, the proposed formula is not useful.

6. Please look for help from English editor to the writing

We have revised our manuscript accordingly, and hope our manuscript meet the standard for publication in PLOS ONE.

Yours sincerely

Jingchun Fan

---

## [Editor Report · Decision Letter 2]

19 Sep 2023

The impact of wearing facemask on COPD patients: A protocol of a systematic review and meta-analysis

PONE-D-23-09305R2

Dear Dr. Fan,

We’re pleased to inform you that your manuscript has been judged scientifically suitable for publication and will be formally accepted for publication once it meets all outstanding technical requirements.

Kind regards,

Dong Keon Yon, MD, FACAAI, FAAAAI

Academic Editor

PLOS ONE

Additional Editor Comments (optional):

This is an excellent paper.
---

## [Editor Report · Acceptance letter]

21 Sep 2023

PONE-D-23-09305R2 

The impact of wearing facemask on COPD patients: A protocol of a systematic review and meta-analysis 

Dear Dr. Fan:

I'm pleased to inform you that your manuscript has been deemed suitable for publication in PLOS ONE. Congratulations! Your manuscript is now with our production department. 

Kind regards, 

on behalf of

Dr. Dong Keon Yon 

Academic Editor

PLOS ONE